# Modified Rotating Reel for Malaxer Machines: Assessment of Rheological Characteristics, Energy Consumption, Temperature Profile, and Virgin Olive Oil Quality

**DOI:** 10.3390/foods9060813

**Published:** 2020-06-20

**Authors:** Biagio Bianchi, Antonia Tamborrino, Ferruccio Giametta, Giacomo Squeo, Graziana Difonzo, Pasquale Catalano

**Affiliations:** 1Department of Agricultural and Environmental Science, University of Bari Aldo Moro, Via Amendola 165/A, 70126 Bari, Italy; biagio.bianchi@uniba.it; 2Department of Agriculture, Environment and Food, University of Molise, Via De Sanctis. n.c., 86100 Campobasso, Italy; ferruccio.giametta@unimol.it (F.G.); catalano@unimol.it (P.C.); 3Department of Soil, Plant and Food Sciences, Food Science and Technology Section, University of Bari Aldo Moro, Via Amendola, 165/A, 70126 Bari, Italy; graziana.difonzo@uniba.it

**Keywords:** olive oil equipment plant, malaxer machine, rotating reel, temperature, viscosity coefficient, energy consumption, olive oil quality, extraction yield

## Abstract

The properties of food products are the result of changes produced in raw materials as a result of process treatments. In the olive oil extraction process, these changes can be observed as differences in quality, nutritional characteristics, taste, and flavor, and are especially due to the time and temperature of the malaxation phase. These parameters are closely related to the mechanical design of malaxer machines. In this study, a new reel model for malaxer machines was designed. The new model was incorporated into an industrial malaxer machine and experimental tests were carried out to study the effects of two different reel designs (modified and unmodified profile) on the rheological characteristics of olive paste, the energy consumption of the plant, and the temperature profile inside the machine. The main commercial parameters of the produced olive oil were studied, as well as the extraction yield and the extraction efficiency of the plant. The malaxer machine equipped with the modified reel showed better homogenization of the paste, which led to improved heat exchange and rheological properties. The results of this study showed that a specific modification of the rotating reel can improve the performance of the malaxer in terms of improving the viscosity of the paste, 127,157.67 (mPa s^n^) for the malaxer with the modified reel at the beginning of malaxation, reaching a final value of 64,626.00 (mPa s^n^) at the end. The unmodified malaxer showed an initial viscosity coefficient of 133,754.00 (mPa s^n^) and a final value of 111,990.67 (mPa s^n^). This led to a reduction in malaxing times, an increase in the work capacity of the plant, and a reduction in total energy consumption and slowed down the oxidative phenomena responsible for the decrease in the quality of olive oil.

## 1. Introduction

In recent years, ever-increasing attention has been paid to aspects of mechanical engineering in order to produce high quality food products. This involves revising the design of certain specific food processing machines. In particular, in the olive oil extraction process, improvements can be observed in terms of quality parameters, such as nutritional parameters, taste, and flavor, mainly due to the control of the time and temperature of the malaxation process [1,2,3,4,5]. These parameters are closely related to the mechanical design of malaxer machines. The malaxation phase determines the correct/incorrect balance between the quality and quantity of oil extracted, depending on various operating parameters: time and temperature as mentioned above, atmosphere near the olive paste, addition of hot water, and use of adjuvants [6,7,8,9,10,11]. The influence of process parameters on quality and oil yield has been widely studied by many authors. Some of these studies have deepened the role of oxygen during the malaxation process by investigating the qualitative and quantitative value of oil and the rheological characteristics of olive paste through the proper design and construction of a pilot plant [11,12,13,14,15,16,17,18,19,20,21]. The design aspects of current machines in the oil industry have also been studied in order to adopt innovative solutions and improve the efficiency of the heat exchange. A circular section of the tank instead of the standard semi-cylindrical section, new sets of blades mounted on the mixing shaft, and a heat exchanger between the crusher and the malaxer have been studied [13,14,15,16,17]. In this way, malaxation has been optimized: reducing malaxation times and increasing the correct final temperature of the paste, consequently reducing the degradation of olive oil and limiting oxidation phenomena. In 2015, Ayr et al. [10] used a 3D thermofluid dynamic model as a first approach to analyze the influence of the presence of two groups of malaxer blades instead of a single set on thermal performance. In this work, some preliminary tests relating to only the temperature trend in a specific point of the machine were performed. The results showed that, in a malaxer with a reel with shorter blades in addition to the standard long blades, the olive paste was better mixed. This solution reduced the non-homogeneity of the temperature in those sections far from the walls of the tank, leading both to a better thermal profile of the olive paste under malaxation and to a reduction in processing times. In fact, the reduction of malaxation times is one of the main objectives of the research in this sector because of its influence on many quality factors and on the optimization of the process management. On the basis of these considerations, several studies have been carried out on the effect on malaxation times of heating olive paste by means of a heat exchanger between the crusher and the malaxer [21,22,23,24,25]. However, there are have been no further studies that simultaneously analyzed the variation of rheological and energy features, in addition to the temperature trend. The preheating of the olive paste has been allowed to reduce the time of malaxation by safeguarding the yield and increasing the quality of olive oil, especially in terms of higher content of volatile components and antioxidants. More recently, research has been carried out to study the introduction of microwaves, ultrasounds, and electric pulsed fields in the olive oil extraction process to improve the thermal and non-thermal effects [26,27,28,29,30,31,32,33,34,35]. Some industrial experiments have been carried out by applying these innovative technologies to the olive oil extraction process, obtaining good results, especially in terms of reducing the total process time [26,27,28]. The use of microwaves is characterized by oil yields comparable to the standard process and by a substantial reduction in the paste processing time, from 40 min (on average) to a few seconds [26]. Despite the importance of a more in-depth research on these aspects, the scientific literature, with the exception of the few publications mentioned above, lacks studies on mechanical solutions aimed at improving the fluid dynamics of the paste in the malaxer and specific evaluations about the effects on the quality and extraction yields. Following the results reported by Ayr et al. in 2015 [10], where the tests were made on a scaled industrial prototype of a malaxer machine, a different reel was specifically designed and studied in this research. In particular, this paper shows the first results of experimental tests carried out in a two-phase, full-scale plant to study the functional and qualitative performance of the modified malaxer machine equipped with the suitably modified rotating reel. The reel profile was specifically designed by adding transverse blades to create a reel profile different from that of the malaxers commonly used in industrial plants.

## 2. Materials and Methods

### 2.1. Industrial Olive Oil Extraction Plant

The experimental tests were carried out in an industrial oil mill (Oleificio Schinosa) located in Trani (Puglia, Italy). The plant is equipped with a hammer crusher (model A60; Amenduni Nicola Spa) and two mixing groups (mixing group model Italia; Amenduni Nicola Spa), each consisting of three horizontal tanks (malaxer) of 1000 kg each. The mixing unit consists of a longitudinal steel shaft connected to a 5.5 kW electric motor with a gearbox suitable for obtaining a shaft rotation speed of 15 rpm and equipped with a malaxer reel. The system is completed by a decanter (model REX 250; Amenduni Nicola Spa) of 3000–8000 kg h^−1^ as the theoretical mass flow rate. The oil is cleaned with a vertical centrifuge (separator A-3500; Amenduni Nicola Spa).

### 2.2. Geometrical Characteristics of the Malaxer Reel

In this section, the concepts and rationales of the proposed modification are presented. The malaxer machine is a kneading machine that receives the olive paste that is produced by breaking olives through milling or crushing. Inside the malaxer machine, the olive paste is kneaded and heated before the separation of the oil from water and husk by centrifugation. During the malaxing process, the olive paste must reach a sufficient temperature to allow disruption of the water–oil emulsion during malaxing and to promote the coalescence phenomena. However, the malaxation time has been recognized as a main factor that causes negative effects on the olive oil quality if it is too long [10,11,12,13,14,15,16]. The time in which the kneading temperature is reached depends on the action and type of the mechanical parts of the malaxer reel and this has effects on the rheological characteristics of the olive paste as well as on the extraction yield and olive oil quality [16]. The temperature reached also determines various physiochemical changes and enzymatic reactions that can occur during this phase, influencing the olive oil quality and extraction yield [1,19]. Industrial malaxer machines are characterized by a malaxer tank with kneading tools made by a reel with different geometry of the blades whose movement in the olive paste leads to temperature distribution. The result can be an inconsistent and non-uniform heat treatment in different areas of the olive paste. To overcome these problems, this study has made efforts to optimize the geometries of the kneading tools (reel).

The common reel (unmodified) consists of an external screw whose profile is as long as the entire tank and is welded to the ends of radial elements that start from the shaft and are suitably inclined (Figure 1). It is quite different from that used in the prototype studied by Ayr et al. 2015 [10] as some transversal elements were added to the reel: the blades were not bidimensional, but they had a little lateral flag to increase mixing effects and more holes to reduce energy consumption. In the study carried out by Ayr et al. [10], a new blade order was added to the reel. As highlighted in the Ayr et al. study, despite good performance of the machine, the effects were still not completely satisfactory. The proposed idea was to increase the mixing action of the malaxer machine by changing differently the blades of the reel. The new modified reel was designed to be even more efficient, thanks to the addition of rectangular elements mounted transversely to about half of the radial elements and suitably inclined (Figure 2). Then, a radial element for each blade was welded. The radial element had a surface of 327 mm length and 79 mm height, and a side strap for solid connection with the blade. All the mechanical parts were made using stainless steel AISI 316L. This extension was intended to improve the kneading of the olive paste, as verified in this paper. The new prototype was tested in an industrial-scale olive oil mill. This machine was new as there was no other industrial malaxer with the same or better characteristics. This represented a new, full-scale prototype and in the next phase, it will be engineered to realize a new, full-scale industrial machine.

### 2.3. Experimental Design

The experimental tests were carried out during the 2018–2019 olive oil season, using a homogeneous batch of olives of about 25,000 kg. The olives (*Olea europaea* L.) of the Coratina cultivar, harvested mechanically, were milled within 24 h. The operating parameters of the hammer crusher were 1500 rpm, 7000 kg h^−1^ as processing capacity, and a grid with 6 mm holes. Two adjacent malaxing groups were used for the comparative tests, one equipped with the “unmodified reel” and the other with the “modified reel”. The decanter operated at 2800 rpm (tank) and 2825 rpm (auger) with a flow rate of 6000 kg h^−1^, without the addition of water. The centrifugal separator operated at 6400 rpm and 2000 L h^−1^. In each test, the malaxation time was 60 min: 50 min of actual time, 5 min of filling time, and 5 min of unloading time. Three industrial repetitions were made for each thesis studied—unmodified reel and modified reel.

### 2.4. Experimental Measurements

#### 2.4.1. Olive Paste Temperature Measurements

The temperature of the olive paste was measured inside the malaxer using a stainless-steel bar with probes (Pt-100 thermocouples ± 0.5 °C) and connected to a multichannel BABUC/A/M data acquisition system (Figure 3). The temperature was measurement in three different points (top layer, middle layer, and bottom layer) for all experimental thesis carried out.

#### 2.4.2. Electric Energy Consumption Measurements

The active electrical power absorbed by the motor of each malaxer was measured by means of a power quality meter and analyzer *CW 121* (Yokogawa Electric Corporation, Tokyo, Japan). Measurements of energy consumption were made when the shaft was already rotating and under stable conditions, starting with the filling phase of the tested malaxer. Measurements were made by inserting the instrument leads into the power line between the electrical panel and the motor of the line under consideration.

#### 2.4.3. Olive Paste Viscosity Measurements

A Brookfield rotational viscometer (model DV2-HBT; Brookfield Engineering Laboratories, Inc., Stoughton, MA, USA) equipped with interchangeable disc spindles 2–7 (model RV/HA/HB; Brookfield DVII + Brookfield Engineering Laboratories) was used for the rheological analysis of olive paste samples. The viscosity measurement was carried out using 600 mL of olive paste, loaded into 1000 mL glass containers conditioned at 27 °C in a thermostatic bath. The apparent viscosity of each sample was recorded at 10 rotational speeds from 0.5 to 100 rpm using the RV/HA/HB-4 spindle. To interpret the experimental results in terms of viscosity, the torque-speed data and scale readings were converted to shear stress and shear speed ratios using numerical conversion values. An empirical power-law model was used to calculate the apparent viscosity and flow behavior index from the shear rate, using the power law equation *η*_app_ = k*γ^n^*^−1^, where *η*_app_ is the apparent viscosity, *γ* is the shear rate (s^−1^), *n* is the flow behavior index (without size), and k is the consistency index (mPa s^n^). The results were expressed as the mean of three replications.

#### 2.4.4. Sampling

The following samples were taken during each test (*n* = 3) and the experimental schedule is shown in Figure 4.

-One kilogram of olives of the processed lot.-Sampling of the paste during the malaxation process was carried out every 10 min from the end of the filling phase to the beginning of the unloading phase. For each sampling, three samples were taken at different levels of the tank (top layer, middle layer, and bottom layer) using a specially constructed sampler (Figure 4).-Three samples of pomace coming from the decanter outlet (at the beginning, at the centre, and at the end of the extraction process) and then homogenized to form a single sample of 1 kg.-Three 100 mL oil samples of the centrifugal separator every 3 min.

### 2.5. Analytical Determinations

#### 2.5.1. Extraction Efficiency of the Plant

The extraction efficiency of the plant was calculated using the following equation:$$\mathrm{EE}=\frac{m_{oil}}{m_{total\;oil}}\times 100$$
where *m_oil_* is the mass of the extracted oil (kg) and *m_total oil_* is the mass of the total oil (kg) content in the processed olives determined as described in the following section. The samples of olive paste were repeatedly collected every 10 min until the beginning of discharge (occurred after about 50 min of malaxation process), representing a total of six sampling times, including sampling at time zero. For each sampling time, three samples were taken at different levels of the tank: top layer, middle layer, and bottom layer. For each sample analyzed, the percentage differences at different levels and at different malaxation times were calculated around the mean value.

#### 2.5.2. Oil Content in the Olives, Olive Paste, and Pomace

The total oil content was determined on 30 g of sample that was dehydrated until reaching a constant weight. The olives sample was also previously milled. The sample was extracted with hexane in an automatic extractor (Randall 148; Velp Scientifica, Milano, Italy) following the analytical technique described by Cherubini et al. [36]. The sample was initially subjected to an immersion phase at 139 °C for 60 min, and the porous container of the sample was immersed directly in the boiling solvent. The sample was then subjected to washing at 139 °C for 40 min, and the sample container was removed from the solvent and reflux-washed. The final part of the process was solvent recovery, which was conducted at 139 °C for 30 min. The results were expressed as the percentage of oil on wet and dry matter.

#### 2.5.3. Olive Oil Analysis

The determinations of free fatty acids (FFAs), peroxide value (PV), and spectrophotometric constants (K_232_, K_268_, ΔK) were carried out according to Regulation EEC 2568/91 and subsequent amendments and integrations [37]. The RapidOxy analysis was conducted as reported by Difonzo et al. [38].

### 2.6. Statistical Analysis

Results are expressed as mean ± SD of three experiments for analytical determination and the consistency coefficient (k). The analysis of variance (ANOVA), followed by the test of the honestly significant difference test of Tukey (HSD) for multiple comparisons, was performed on the experimental data using the XLStat software (Addinsoft SARL, New York, NY, USA). *p* ≤ 0.05 was considered to be statistically significant.

## 3. Results and Discussion

### 3.1. Functional Performance of the Different Models of the Reel: Evaluation of Extraction Efficiency

The mean value of fat content of olive paste was 11.71%, calculated on a wet basis. Using this value as a reference, Table 1 was constructed to show the percentage differences at different levels and at different malaxation times around the abovementioned mean value. At the start of each test, after 5 min of filling, the paste was already stratified in terms of oil content in the malaxer. More precisely, in the unmodified malaxer, the paste had a higher oil content (3.52%) in the bottom layer than in the middle (−4.36%). Subsequently, the paste, which was first placed in the tank and mixed longer during the filling phase, accumulated more oil, but remained concentrated at the bottom. On the contrary, always in the initial moment, in the modified kneading machine, there was a greater uniformity of the oil content in the layers of paste with a relative variation of less than 1.5% compared with the average value. After 10 min of malaxation, the difference in oil in the layers of paste continued to be noticed in the unmodified malaxer, even though the lower layer was now better mixed with the middle layer because both had a similar percentage of oil (Table 1).

On the other hand, the modified malaxer was again better mixed. After 20 min of malaxation, the layer with the highest oil content was the central layer in the unmodified malaxer and the top layer in the modified malaxer. In the latter case, a less significant difference in oil content between the layers was seen (Table 1). After 30 min, the layer with the highest oil content was on the surface in the unmodified malaxer. Unlike the other samples, in the modified malaxer, there was a significant but small difference between the percentage of oil in the lower layer and the other layers. During the last 20 min of malaxation, the unmodified malaxer continued to behave with a high difference in oil content between layers, with a higher concentration of oil in the lower layer, while the modified malaxer tended to behave uniformly throughout the mass of paste. In summary, in the unmodified malaxer, there was a significant lack of homogeneity in the distribution of fat throughout the malaxation time in accordance with the speed profiles found in Ayr et al. (2015) [10], while in the modified malaxer, there was substantial uniformity in the distribution of fat linked to the action of the added elements, especially after 40 min of malaxation. It was therefore evident that the changes on the reel allowed a better homogenization of the paste and also a more effective separation of the fat phase at the same time of malaxation, without creating differences in the oil content in the different layers of paste sent to the decanter. This aspect has proved to be very important to maintain a good oil yield. In fact, the decanter is still a rigid machine and allows maximum separation efficiency if fed with homogeneous and adequately prepared paste, especially if it works by discharging only two phases (oil must and pomace). If the oil content and rheological characteristics of the olive paste that feeds the decanter vary over time, the machine must be continuously and rapidly adjusted in its operating parameters (for example the ΔN or the thickness of the oil level). These results also explained the difference in the oily residue in the pomace coming from the paste obtained with the unmodified malaxer (17.1%) and that of the pomace coming from the paste obtained with the modified malaxer (6.7%). It was evident that the decanter was strongly influenced by the less homogeneous characteristics of the upper layer of the paste in the unmodified malaxer, providing a lower overall extraction efficiency.

### 3.2. Temperature Profile Evaluation

Figure 5 shows the distribution of the temperature in the tank, during the malaxation time, at different levels.

In the malaxer with an unmodified reel, the middle layer of the olive paste tended to take on lower temperatures than those of the upper and lower layers. After 10 min of malaxation, the top layer had the highest temperature and there was a difference of 1.1 °C between the temperature of the top layer and that of the bottom layer, while there was a difference of 3.3 °C between the temperature of the top layer and that of the middle layer. After 30 min, the temperatures in the three layers were very similar, but after 40 min, there was still a difference of about 2 °C between the top layer and the middle layer. At the end of malaxation, after 50 min, the layer with the highest temperature was the lowest, the layer with the lowest temperature was the highest, and there was a temperature difference of 2.5 °C between the two layers. In any case, the temperature in the unmodified malaxer was always below 26.5 °C (Figure 5). In the modified malaxer, a more regular distribution of temperatures was seen. However, the temperature tended to be slightly higher in the floor layer in less than 10 min of malaxation; at this moment, a temperature difference of 2.4 °C between the lower and central layers was noticeable. The temperature difference between the top and middle layers after 40 min was in the same order of magnitude: 2.5 °C. At the end of malaxation, the temperatures in the lower and middle layers were almost equal and were about 1.9 °C lower than those in the upper layer (Figure 5). In the unmodified malaxer, it was clear that the warmest layer at the end of malaxation was the lower one, which was more in contact with the heated walls of the tank and where the worst heat transfer by convection occurred because of the worst mixing of the layers. The malaxer with the modified reel produced, at the end of malaxation, a slightly more effective thermal mixing (Figure 5).

### 3.3. Energy Consumption Evaluation

Figure 6 shows the trend over time of the active electric power absorbed by the kneading motor during the experimental tests. It should be noted that the insertion of transversal blades on the radial elements and the variations in viscosity of the olive paste during the malaxation process did not significantly influence the energy consumption of the malaxer during the entire process, excluding, of course, the filling and unloading phases. In fact, the trend in energy consumption was the same in the two tests. During the filling phase, the malaxer motor absorbed power that gradually increased from 2.30 kW to 2.60 kW (mean value of the samples acquired during the malaxation phase), while during the unloading of the tank, the absorbed power dropped from the speed value to about 2.1 kW (Figure 6). During the regime phase, the active power absorbed by the machine settled around the average value between 2.50 kW and 2.55 kW in both tests, oscillating between 2.40 kW and 2.60 kW for the unmodified malaxer and between 2.50 kW and 2.60 kW for the modified kneading machine, with no significant differences between the regime means. Therefore, the characteristics of the olive paste and the mechanical solutions adopted did not change the resistant torque on the rotating reel blades, so at constant angular speed, the energy consumption remained almost constant. The slightly higher power absorbed by the modified machine suggested that the absorption of the active power of the motor was influenced by a small increase in resistance due to the added elements and was not or little influenced by the lower resistance due to the reduction in the viscosity of the paste. Finally, it must be considered that the proposed mechanical solution, with the same oil yield, could allow the reduction of mixing times, obtaining an increase in the operating capacity of the plant and a significant reduction in total energy consumption.

### 3.4. Viscosity Evaluation

The trend of the viscosity coefficient of the olive paste, from the beginning of malaxation to discharge into the decanter (Figure 7), tended to confirm the results obtained.

The paste at the beginning of malaxation had substantially similar values of coefficient of viscosity, 133,754.00 (mPa s^n^) for the malaxer with unmodified reel and 127,157.67 (mPa s^n^) for the malaxer with the modified reel. The viscosity coefficient of the paste in the malaxer with unmodified reel alternated between decreasing and increasing values: up to 115,772.67 (mPa s^n^) after 10 min and from 146,271.67 (mPa s^n^) to 103,481.33 (mPa s^n^) between 30 and 40 min, reaching a final value of 111,990.67 (mPa s^n^) (Figure 7). This confirmed the abovementioned observation that the unmodified reel did not behave uniformly over time with regard to fat distribution and temperatures. On the contrary, the viscosity coefficient of the paste in the malaxer with the modified reel continuously decreased over time without maximum or minimum values, until it reached the lowest value at the end of malaxation: 64,626.00 (mPa s^n^) at 50 min (Figure 7).

This final value was also lower than the final value obtained in the unmodified malaxer. In fact, the unmodified malaxer was characterized by poor variations of the viscosity coefficient during different times—mean values did not differ significantly—the global mean value of 122,800 (mPa s^n^) was significantly similar to that observed in the first malaxation phase (average of 122,400 (mPa s^n^) for the first 30 min) and was significantly different from the final values (average of 71,700 (mPa s^n^) for the last 20 min). This was the second reason that explained the lower oil content in the pomace coming from the paste obtained with the modified malaxer instead of the unmodified one. The above discussion allowed us to affirm that the insertion of transversal elements corresponding to the central line of each radial element had a positive mechanical effect on the paste, also allowing better breakage of water–oil emulsions, with better separation of the oil from the solid and liquid phases leading to a lower coefficient of viscosity.

### 3.5. Olive Oil Quality Results

The performance of the modified and unmodified malaxer machines was also compared in terms of product quality and the results are summarized in Table 2. Both the oils belonged to the extra virgin olive oil (EVOO) commercial class, based on the measured parameters, as established by EEC 2568/91 and subsequent modifications and integrations, confirming the good quality of the processed olives. However, significant differences were observed due to the implemented machine modifications.

Firstly, a significant difference in FFAs was observed, which increased in the modified malaxer with respect to the unmodified one. Free fatty acids are set free as a result of lipase (triacylglycerol acylhydrolase; EC 3.1.1.3) activity in the presence of water, which actually occurs only when the drupes have been damaged on the tree or at the olive mill [39]. In our work, the higher extention of hydrolitic degradation in the modified malaxer may be considered as the “other side of the coin” of the improved efficiency in mixing. Indeed, the better distribution of the whole oily phase throughout the machine layers (Table 1) might result in a less mass/surface ratio at each point and thus in a closer contact of such phase (oily) with water and enzymes. It has been proven that, during mixing, lipase is active and may increase the amount of free fatty acids [40].

Although no significant differences were observed in the PV, which in any case was generally high when considering freshly produced oils [8], a different oxidative profile of the oils was also depicted; the modified malaxer generally gave more oxidized oils than the unmodified one.

The value of K_232_, a marker of primary oxidation products, was higher in the oils obtained from the modified malaxer than the control. O_2_ availability in the malaxer chamber induced the initial lipid oxidation reaction, with a consequent increase in conjugated dienes and their oxidation products, which absorb at λ = 232 nm [41]. K_268_ and ΔK did not show significant difference between oils obtained from the modified and unmodified malaxers and this was not surprising, considering that these parameters are secondary oxidation markers. The higher oil oxidation in the modified malaxer might be linked to higher O_2_ availability in the machine chamber or increased exposure of the oily phase to oxygen, due to the lower mass/surface ratio of that phase, as previously outlined for FFA.

These considerations were confirmed by looking at oxidative stability, expressed as induction time and evaluated by RapidOxy. The data described a significantly higher oil oxidative stability in the unmodified malaxer than in the modified one. The lower oxidative stability in the newly designed machine could be related to both the higher extent of primary oxidation and/or higher FFA content. With regard to this last aspect, several papers [42,43] have been published on the pro-oxidant action of FFAs. Autoxidation of oils is accelerated by the presence of free fatty acids; it seems to be exerted by the carboxylic molecular group, which accelerates the rate of decomposition of hydroperoxides [44,45].

## 4. Conclusions

In two-phase virgin olive oil extraction plants, mechanical malaxation must be particularly rapid and efficient, considering that it is still the most widely used solution for industrial production. The processing speed avoids long malaxation times with excessive oxidation of the oil and the efficiency reduces oil loss in the pomace. In fact, in the two-phase decanter, the middle layer of water is thinner than the three-phase decanter and all the unextracted oil flows into the pomace layer. It is known that this oil could be partially recovered with a second extraction decanter, but it would have worse quality characteristics than the oil that is extracted first. Currently, the mixing mechanism of the olive paste consists of a peripheral stainless-steel coil welded to the ends of perforated radial profiles starting from the longitudinal shaft. These radial elements are properly angled with respect to the shaft and its cross section. The results of this experimental study were obtained by using a full-scale industrial plant. They showed that the insertion of rectangular elements, mounted transversely to about half of the radial elements and appropriately inclined with respect to them, represented a mechanical modification that improved both the homogenization of the paste and heat exchange without altering energy consumption. As far as the rheological behavior of the olive paste was concerned, the use of the modified malaxer also improved its viscosity coefficient, which tended to decrease with the processing time. Moreover, the oil could be better separated from the solid and liquid phases, leading to a lower percentage of oil in the pomace. Finally, the quality evaluation of the extracted oil showed that changes in the malaxer reel also led to an intensification of oxidative events. This could be due to higher oxygenation of the olive paste during malaxation, likely due to increased O_2_ dissolution in the olive paste and/or an increased surface/mass ratio of the oily phase which led to a higher oxygen exposure. However, considering the possibility of a significant reduction in the mixing times—thanks to the modified reel—the risks of oil oxygenation will also be lowered. Finally, the possibility of reducing malaxation times would bring advantages both in terms of increasing the operating capacity of the plant and reducing the overall and specific energy consumption.

## Figures and Tables

**Figure 1 foods-09-00813-f001:**
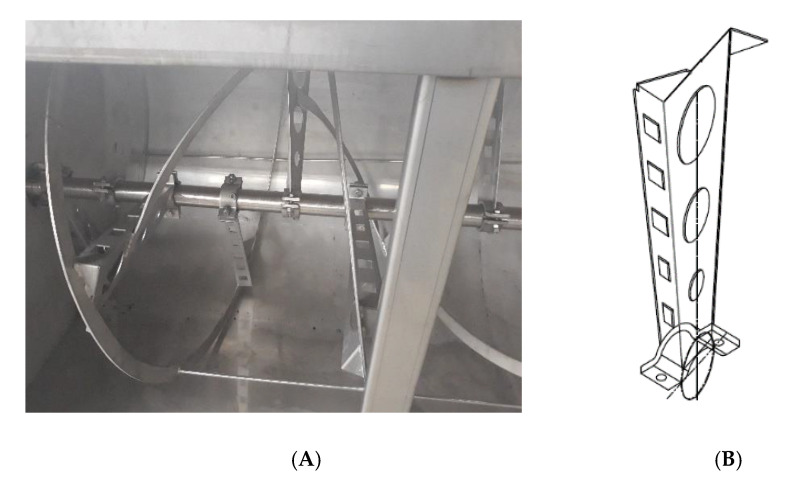
Details of the unmodified reel. (**A**) malaxer tank equipped with unmodified reel and (**B**) isometric view of the unmodified radial element of the reel.

**Figure 2 foods-09-00813-f002:**
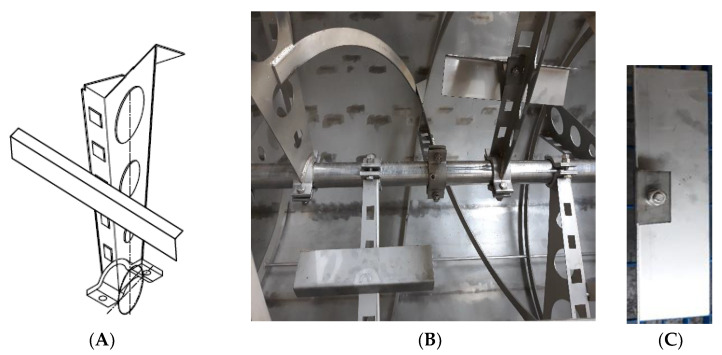
Details of the modified reel (**A**) Isometric view of the modified radial element of the reel, (**B**) malaxer tank equipped with modified reel and (**C**) the rectangular shaker element mounted on the radial elements.

**Figure 3 foods-09-00813-f003:**
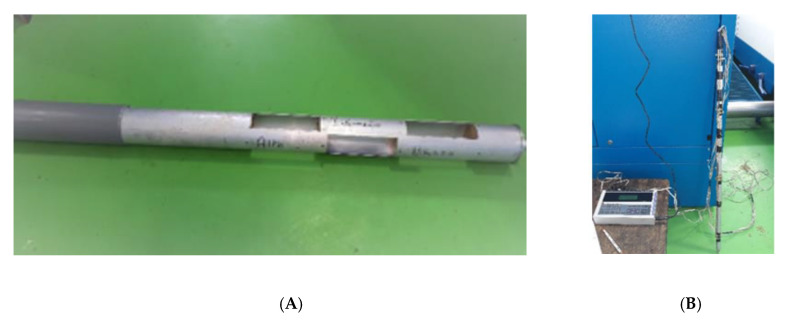
Olive paste sampler and sampling from the malaxer. (**A**) stainless steel bar with probes arranged in three different points and connected (**B**) to the BABUC data acquisition system.

**Figure 4 foods-09-00813-f004:**
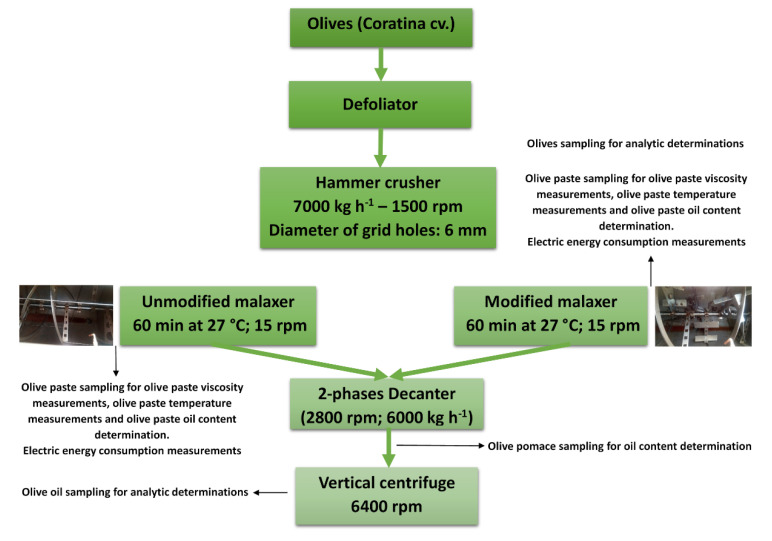
Flow chart of the production process and samples taken.

**Figure 5 foods-09-00813-f005:**
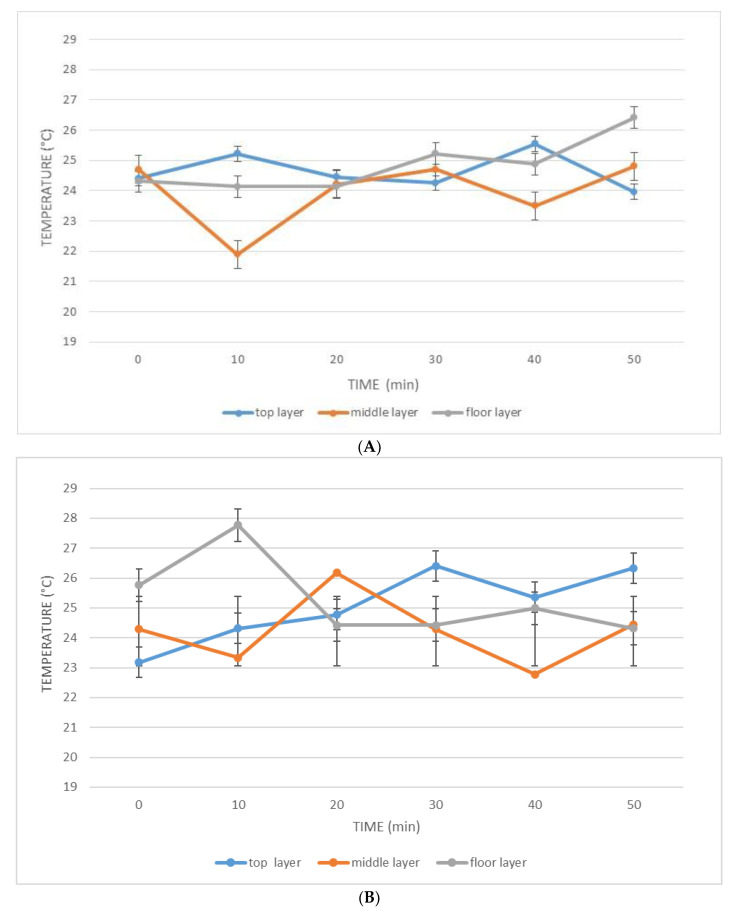
Distribution over time of the temperature inside the (**A**) unmodified malaxer and (**B**) modified malaxer.

**Figure 6 foods-09-00813-f006:**
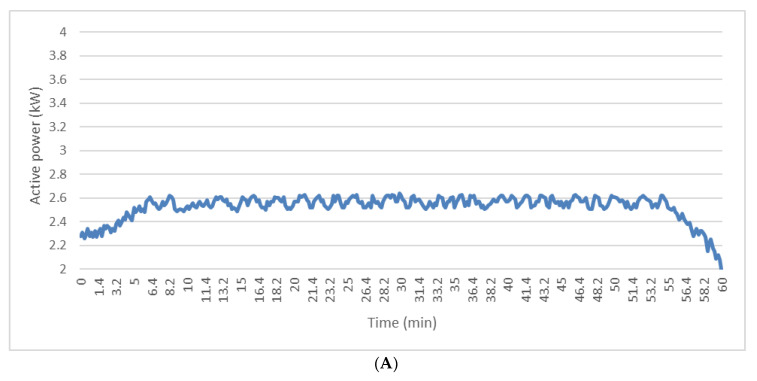
Active electric power absorbed from malaxers as a function of time during the experimental tests. (**A**) Unmodified malaxer and (**B**) modified malaxer.

**Figure 7 foods-09-00813-f007:**
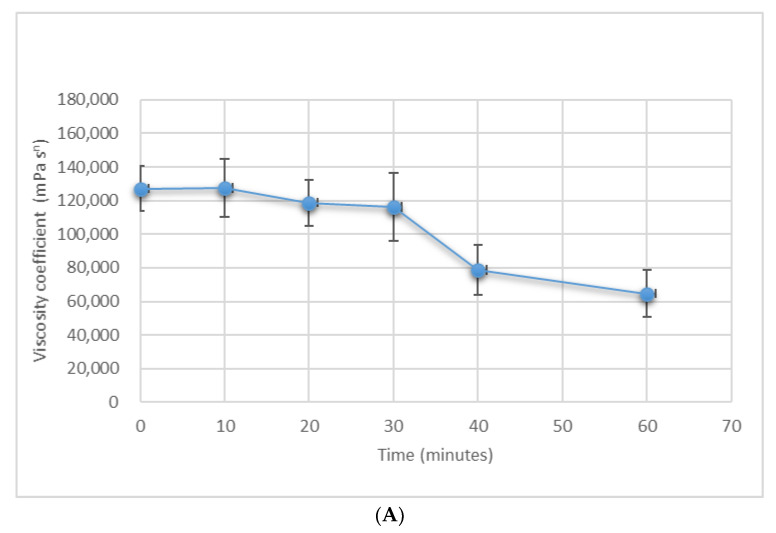
Paste average coefficient of viscosity versus time in the (**A**) unmodified malaxer and (**B**) modified malaxer. Data with the same letter do not significantly differ from each other, *p* ≤ 0.05.

**Table 1 foods-09-00813-t001:** Distribution over time of the separable fat content within the malaxers.

	Type of Blades	Top (%)	Middle (%)	Bottom (%)
0	Unmodified	0.84 ± 0.042 b	−4.36 ± 0.04 b	3.52 ± 0.11 a
	Modified	1.44 ± 0.10 a	0.00 ± 0.00 a	−1.44 ± 0.03 b
10	Unmodified	3.73 ± 0.10 a	−3.13 ± 0.25 b	−0.59 ± 0.01 b
	Modified	−0.60 ± 0.10 b	0.17 ± 0.01 a	0.51 ± 0.03 a
20	Unmodified	−4.29 ± 0.13 b	7.47 ± 0.07 a	−3.09 ± 0.15 b
	Modified	3.66 ± 0.07 a	−3.15 ± 0.19 b	−0.43 ± 0.10 a
30	Unmodified	3.57 ± 0.04 a	3.24 ± 0.10 a	−6.80 ± 0.20 b
	Modified	2.57 ± 0.15 b	−1.68 ± 0.03 b	−0.89 ± 0.02 a
40	Unmodified	1.46 ± 0.12 a	−10.70 ± 0.53 b	9.16 ± 0.09 a
	Modified	0.67 ± 0.03 b	1.10 ± 0.07 a	−1.71 ± 0.10 b
50	Unmodified	−14.35 ± 0.72 b	6.03 ± 0.48 a	8.39 ± 0.67 a
	Modified	−0.37 ± 0.02 a	−1.95 ± 0.08 b	2.43 ± 0.10 b

(Data are the mean ± SD of three experiments; percentage differences of the separable fat content with respect to the mean value on a wet basis. Data with the same letter do not significantly differ from each other, *p* ≤ 0.05).

**Table 2 foods-09-00813-t002:** Quality features of the virgin olive oil produced by the unmodified and modified malaxer machines.

	Unmodified	Modified	*p*-Values
**Free fatty acids (g 100 g^−1^)**	0.33 ± 0.05	0.55 ± 0.09	<0.05
**Peroxide value (mEq O_2_ kg^−1^)**	12.95 ± 0.49	13.77 ± 0.34	0.09
**K_232_**	1.59 ± 0.05	1.80 ± 0.12	<0.05
**K_268_**	0.14 ± 0.01	0.16 ± 0.01	0.31
**ΔK**	0.00	0.00	-
**RapidOxy (minutes)**	112.61 ± 3.12	77.52 ± 4.41	<0.05

Data are the mean ± SD of three experiments (*p* < 0.05 indicates significant differences according to one-way ANOVA followed by Tukey’s test).

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
