# Peer review of "Modified Rotating Reel for Malaxer Machines: Assessment of Rheological Characteristics, Energy Consumption, Temperature Profile, and Virgin Olive Oil Quality"

_foods, 2020, doi:10.3390/foods9060813_

Round 1

Reviewer 1 Report

Authors must clarify to whom the design of the mixer used is attributed, and wether this was designed specifically for the essay presented in the manuscript or not.

The manuscript lacks clarifying what is new in this research study regarding the article referred, by Ayr et al. (2015).

The title refers to 'Modified rotating reel for malaxer machine'. Is it new?

Also it refers to 'assessment of rheological characteristics, energy consumption, temperature profile and virgin olive oil quality'. However it seems to be not new regarding the referred article (Ayr et al., 2015). This ainformation should be clearly transparent.

The whole results and discussions should be revised to report and clarify what is new.

Author Response

Thank you for your interest in our research and for your advice in revising our manuscript.

In order to satisfy clearly the Reviewers' comments, all accepted, we will indicate your comments followed by our revisions.

Authors must clarify to whom the design of the mixer used is attributed, and wether this was designed specifically for the essay presented in the manuscript or not.

The malaxer machine equipped with the prototype of the modified blades was specially designed and studied in this research and the results presented are the first ever achieved. The modified sentence was added in the end of the introduction.

The manuscript lacks clarifying what is new in this research study regarding the article referred, by Ayr et al. (2015).

The article by Ayr et al. (2015) reports on a first experimental design and analysis approach to invest on modification made at the reel of a malaxer validation for prediction of heat transfer in a malaxer machine equipped with the modified rotating reel. In that research the design of the rotating reel involved adding a new order of shovels to the reel. The study of the modifications of the blades of the malaxer machine was taken up by the authors of this manuscript who instead evaluated a new model of mixing blades in which they envisaged the addition of transversal elements. This has been specified in the text.

The title refers to 'Modified rotating reel for malaxer machine'. Is it new?

As sad above it is a new model of rotating reel, designed and build specifically for this research.

Also it refers to 'assessment of rheological characteristics, energy consumption, temperature profile and virgin olive oil quality'. However it seems to be not new regarding the referred article (Ayr et al., 2015). This ainformation should be clearly transparent.

The whole results and discussions should be revised to report and clarify what is new.

In the paper ayr et al, 2015, a 3D simulation was made and comparing it with the results of preliminary experimental tests. Rheological tests, energy consumption and temperature profiles have never been carried out simultaneously on a single machine. In this an industrial machine was specifically modified to improve its performance basing the idea on the very preliminary results of the previous paper.

A sentence has been added that clarifies the contents of the title.

Reviewer 2 Report

The manuscript aimed to see whwter and how Modified rotating reel for malaxer machines affect the rheological characteristics of olive paste, the energy evaluation ot the plant, and the temperature profile inside the machine.   The topic is more than interesting and is of great interest for VOOs producers.

However, there are some minor points needed in order to make the paper appropriate for publication.

Line 28. Delete ‘on’ after first time mentioned (in listing aims)

Line 32 – 36. please add main results (in terms of percentage or numbers or similarly) which are currently descriptively written

Line 46. Dot is missing

Line 103. Rephrase: ‘if it is too long’

Figure 4. At the flow chart include sampling of olive past for viscosity measurements, and other analyses from Modified malaxer

Line 233. P ≤ 0.05, and change trough the text if applicable.

Table 1. In footnotes add explain on which data stat is refer

Fig 5. Instead up and down, use A and B (please apply to other Figures as well). Also harmonize used expressions of the layers, exp. Or middle (fig) or intermediate (text) or central (line 322), bottom vs floor …

Fig 7. Please make stat analysis to see is there any differences during the time. Indeed, there is trend visibly, but is questionably is this trend make a difference

Line 398. For good explanation, this phrase (and generally the whole MS) and to test your hypothesis - positive aspects of modified malaxer checked and compared should be results of measurements of this two malaxer. Thus, please add stat analyses in at least few key phases (20, 40, 60 min) or similar. Same goes for results of energy consumption

After addition it could be discuss in this terms and it could maybe be confirmed stated in conclusion

Line 422-424 – second sentences rephrase: although no significant differences were observed in the PV….

Author Response

The manuscript aimed to see whwter and how Modified rotating reel for malaxer machines affect the rheological characteristics of olive paste, the energy evaluation ot the plant, and the temperature profile inside the machine.   The topic is more than interesting and is of great interest for VOOs producers.

However, there are some minor points needed in order to make the paper appropriate for publication.

Line 28. Delete ‘on’ after first time mentioned (in listing aims)

Done

Line 32 – 36. please add main results (in terms of percentage or numbers or similarly) which are currently descriptively written

Done

Line 46. Dot is missing

Done

Line 103. Rephrase: ‘if it is too long’

Done

Figure 4. At the flow chart include sampling of olive past for viscosity measurements, and other analyses from Modified malaxer

This question is not clear: do we have to delete this information in the flowchart? If so we think it could help the reader in understanding how experimental tests were carried on, but if required we will delete the information.

Line 233. P ≤ 0.05, and change trough the text if applicable.

Done

Table 1. In footnotes add explain on which data stat is refer

Done

Fig 5. Instead up and down, use A and B (please apply to other Figures as well). Also harmonize used expressions of the layers, exp. Or middle (fig) or intermediate (text) or central (line 322), bottom vs floor …

Done

Fig 7. Please make stat analysis to see is there any differences during the time. Indeed, there is trend visibly, but is questionably is this trend make a difference

Done

Line 398. For good explanation, this phrase (and generally the whole MS) and to test your hypothesis - positive aspects of modified malaxer checked and compared should be results of measurements of this two malaxer. Thus, please add stat analyses in at least few key phases (20, 40, 60 min) or similar. Same goes for results of energy consumption

Done. As regards energy consumption, statistical analysis is rarely carried on energy plots. In any case a sentence has been added.

After addition it could be discuss in this terms and it could maybe be confirmed stated in conclusion

Done: a sentence has been added below fig. 7

Line 422-424 – second sentences rephrase: although no significant differences were observed in the PV….

Done

Reviewer 3 Report

This is an interesting manuscript describing the study of a prototype malaxer machine equipped with a suitably modified rotating reel. This is an important issue since the quality and nutritional characteristics of virgin olive oil are closely related to the mechanical design of malaxer machines. The results show that specific modification can improve the performance of the malaxer. It should be interesting to study the influence on the phenolic profile in a future study.

The manuscript is well organized and written with comprehensively described methods, and detailed result analyzes and discussion.

Minor comments:

The results in Table 2 show that modification led to the deceased oxidative stability, and the authors conclude that modification reduces the risk of oxygenation, this should be clarified.

I suggest using the symbol m for mass, instead of w, in line 206.

The authors should add the explanation for letters in Table 1, and express result with the same number of figures throughout the table.

Author Response

This is an interesting manuscript describing the study of a prototype malaxer machine equipped with a suitably modified rotating reel. This is an important issue since the quality and nutritional characteristics of virgin olive oil are closely related to the mechanical design of malaxer machines. The results show that specific modification can improve the performance of the malaxer. It should be interesting to study the influence on the phenolic profile in a future study.

The manuscript is well organized and written with comprehensively described methods, and detailed result analyzes and discussion.

Minor comments:

The results in Table 2 show that modification led to the deceased oxidative stability, and the authors conclude that modification reduces the risk of oxygenation, this should be clarified.

Sentence added in the text.

I suggest using the symbol m for mass, instead of w, in line 206.

Done

The authors should add the explanation for letters in Table 1, and express result with the same number of figures throughout the table.

Done

Round 2

Reviewer 1 Report

The manuscript still lacks clarifying the most important concept. Authors added a statement as follows:

'The reel profile was specifically designed by adding transverse 93 blades to create a reel profile different from that of the malaxers commonly used in industrial plants.'

However, surprisingly, they fail to explain what is different regarding the design previously reported by Ayr et al. 2015. Because of this, it is not clear the manuscript provides new information. 
